# Automated Lung Cancer Segmentation in Tissue Micro Array Analysis Histopathological Images Using a Prototype of Computer-Assisted Diagnosis

**DOI:** 10.3390/jpm13030388

**Published:** 2023-02-23

**Authors:** DaifAllah D. Althubaity, Faisal Fahad Alotaibi, Abdalla Mohamed Ahmed Osman, Mugahed Ali Al-khadher, Yahya Hussein Ahmed Abdalla, Sadeq Abdo Alwesabi, Elsadig Eltaher Hamed Abdulrahman, Maram Abdulkhalek Alhemairy

**Affiliations:** 1Pediatric Nursing Department, Faculty of Nursing, Najran University, Najran 66441, Saudi Arabia; 2Strategy Studies and Planning Department, Prince Sultan Medical Military City, Riyadh 13521, Saudi Arabia; 3Community and Mental Health, College of Nursing, Najran University, Najran 66441, Saudi Arabia; 4Nursing College, Najran University, Najran 66441, Saudi Arabia

**Keywords:** lung cancer, automatic identification, TMA, CAD, Tumor, histopathological images

## Abstract

Background: Lung cancer is a fatal disease that kills approximately 85% of those diagnosed with it. In recent years, advances in medical imaging have greatly improved the acquisition, storage, and visualization of various pathologies, making it a necessary component in medicine today. Objective: Develop a computer-aided diagnostic system to detect lung cancer early by segmenting tumor and non-tumor tissue on Tissue Micro Array Analysis (TMA) histopathological images. Method: The prototype computer-aided diagnostic system was developed to segment tumor areas, non-tumor areas, and fundus on TMA histopathological images. Results: The system achieved an average accuracy of 83.4% and an F-measurement of 84.4% in segmenting tumor and non-tumor tissue. Conclusion: The computer-aided diagnostic system provides a second diagnostic opinion to specialists, allowing for more precise diagnoses and more appropriate treatments for lung cancer.

## 1. Introduction

Lung cancer is a malignant tumor that consists of the abnormal growth of both lung and bronchial cells. Multiple factors increase the probability of developing lung cancer, the most important being to bacco use since it is related to around 90% of lung tumors [1]. Lung cancer is a lethal disease that is the fifth cause of death worldwide, the third in Europe, and the first in Iraq [2,3], with almost 20,000 new cases each year; approximately 85% of subjects with lung cancer will die from this disease. The main obstacle in the fight against this pathology is its late detection [4].

The development that the field of medical imaging has undergone in aspects such as acquisition, storage, and visualization has contributed to improving the quality of the analysis and diagnosis of different pathologies (among them lung cancer), making it an essential component in medicine [5]. In recent decades, numerous efforts have been made to detect lung cancer early through the development of different technologies, including computer-aided diagnosis (CAD) systems, which, through the automatic analysis of medical images, provide the specialist with a second diagnostic opinion with the aim of obtaining more precise diagnoses that allow for formulating more appropriate treatments [6].

TMA is a very efficient technique for the analysis of histological tissues; it is currently widely used in the detection of multiple types of cancer, including lung cancer [7] and CAD is generally defined as “systems that perform a quantitative analysis on the medical image and its result is taken into account by a specialist when making a diagnosis [8]”.

Today the only definitive method to confirm the presence of most types of cancer (including lung) as well as to classify them is, according to Terán et al. (2014) [9], a histopathological analysis of a biopsy or tissue sample, since this allows to clearly visualize the state of the disease in addition to helping with the formulation of more appropriate treatment; in this sense, various investigations have been carried out on CAD systems for the histopathological image analysis [10]. The objectives in this aspect are ambitious with attempts to create systems that help with the detection and classification of the disease but also to quantify it, due to the need to know more precisely the state of cancer in order to improve its estimation and predict your progress. One aspect to differentiate is that in radiology, according to Das et al. [11], they have constructed CADs to detect and characterize cancer; however, in histopathology the simple identification of the presence or absence of cancer or even its exact spatial extension may not be of as much interest with respect to more sophisticated questions such as the grade of cancer. Also, on a histological scale, one can begin to distinguish between different histological subtypes of cancer, which is absolutely impossible (or at least difficult) on the radiological scale considering the inherent limitation in the spatial resolution of radiological data. Histopathological medical imaging is the “gold standard” in the early detection of most diseases, including lung cancer [12]. The detection task is usually quite tedious and involves a significant investment of time and effort by histopathology experts. Shazia et al. [2] say that the growth of tissue banks has already outpaced available manual analysis skills, and expert pathology review suffers from inter- and intra-observer variation, highlighting the need to automate the analysis of medical imaging in histopathology [13]. Unlike radiology where the images are in grayscale, in histopathology images are obtained in color. In addition, the data sets are considerably larger and more diverse. For example, a high-resolution chest X-ray has about 134 million voxels while a prostate biopsy could contain about 4 billion pixels among its samples, and even its digitization and storage have been difficult challenges to overcome [14]. These limitations, which in the past might have prevented the advancement of CAD in histopathology, are now no longer obstacles to progress made in digital imaging, storage, and Picture Archiving and Transmission System (*PACS)* systems. The need to create computer-assisted diagnostic systems that take advantage of the advances made in histopathological medical imaging for the early detection of different diseases is urgent. The contributions of the study are as follows:The study aimed to develop a computer-aided diagnostic system to detect lung cancer early by segmenting tumor and non-tumor tissue on Tissue Micro Array Analysis (TMA) histopathological images.Histopathological medical imaging is the “gold standard” in the early detection of most diseases, including lung cancer.The study compares three image enhancement techniques with histogram equalization showing significant improvement in final segmentation of the CAD system.The classification algorithm that presented the best performance for the three data sets of the three image improvement transformations was Random Forest.

### Review of Literature

In an effort to obtain patient survival data, methods have been developed that, in addition to detecting cancer, can obtain indicators associated with patient survival and life prognosis. The first work done by Kun-Hsing et al. [15] focuses on lung cancer diagnosis. Here is a comparison of classifiers among which Random Forest obtains the best result with an area under the curve (AUC) measurement of 0.85. In the second work Mejia, H. et al. [16] a method for breast cancer detection based on pre-segmentation by super pixels is proposed. First-order statistical type characteristics are extracted, then a regulated logistic regression classifier that achieves an accuracy of 89% in segmentation is added. Both studies use data sets from the Stanford TMA database. The quantification of cancer is essential in this type of study. It is very interesting that segmentations are used to obtain patient survival data; this is an extremely important aspect and is taken into account in the framework of this work.

In this work, an approach is made to the detection of lung cancer in medical imaging, specifically addressing the problem of segmentation of tumor and non-tumor tissue on TMA histopathological images through the development of a prototype system of CAD computer-aided diagnosis.

## 2. Materials

### 2.1. The Data Set

The data set has been built in collaboration with the CIMA (Center for Applied Medical Research) in Pamplona, which provided a total of 10 TMA images of different adenocarcinoma-type lung tumors with an immunohistochemical (IHC) stain for the RRM2 marker along with the 328 TMA images out of 205,161 images from the Stanford TMA database [17]. These images were manually marked by experts using ImageJ software. It is worth mentioning that to make a comparison with related work, a search was made for similar data sets. However, none with the necessary requirements were found (essentially images marked by experts) since, although there are initiatives focused on the conformation of open databases of TMA images such as the Stanford TMA database, it is somewhat difficult to find the respective markup of the image by the expert.

#### 2.1.1. TMA Images

The provided TMA images look similar to the one shown in Figure 1. The image is square, with the fabric in the form of a circle differentiated from the light gray background, with an approximate size of 6000 × 6000 pixels in JPEG format. On the other hand, each image corresponds to a file with the marking corresponding to the tumor areas in ROI format, made with the ImageJ program using its polygon selection tool.

#### 2.1.2. Labeled TMA Images

The objective proposed by the experts was to differentiate three regions of interest: the background, the tumor zone, and the non-tumor zone. Initially, the images only marked the tumor area, so it was necessary to create images marked with the three zones called “label images”. Initially, this task seemed trivial since having marked the tumor area (which is the most important), it would only be possible to mark the non-tumor area and the background; however, to try to avoid subjectivity, it was decided to automate the task with the processing program of GIMP images using its fuzzy selection tool: the label 0 (white) was assigned to the background area, then to the tumor area, as already marked; the label 2 (red) was assigned, finally leaving the non-tumor area that was selected by making an exclusion with the first zones, assigning the label 3 (green color). This procedure could be carried out in many ways. Therefore, no further details will be given. The labeling result for one of the original TMA images is shown in Figure 2.

It is important to mention that the tagged images were reviewed in search of possible errors in the marking, thus ensuring consistency with their respective areas. Finally, the labeled images were converted to 8 bits with the intention that only the 3 values corresponding to the labels were found when reviewing their histogram.

### 2.2. The Software

The implementation of the proposed method was carried out in the hands of two major image processing platforms. These are the FIJI ImageJ version 1.53t distribution used for image pre-processing and KNIME for feature extraction tasks including classification and evaluation of the model. Both programs are free, multiplatform, well-known, and widespread. This is precisely the reason for its choice to facilitate the experiment’s reproducibility before the scientific community.

#### KNIME Analytics Platform

Using the Konstanz Information Miner (KNIME), data streams can be visually assembled and interactively executed in a simple and intuitive manner. As a teaching, research, and collaboration tool, it is easy to add new modules or nodes that can incorporate new algorithms and tools, as well as data manipulation and visualization approaches [18,19]. An interactive study of analysis results or trained models is possible through its user interface, which is both powerful and intuitive. The workflow is basically made up of nodes (representing the actions) and arrows (representing the data flow) that are combined and executed interactively, and can be easily observed and controlled. One of the most relevant aspects of KNIME is its high capacity for integration with art tools such as Weka, Matlab, ImageJ, R, and Python. Collections of nodes, known as extensions, currently exist for various fields such as image processing, data mining, business intelligence, financial analysis, and chemical computing.

KNIME Image Processing Extension: KNIME has an image processing plugin that contains about 100 nodes to deal with various types of images (2D and 3D) and videos, in addition to applying common methods such as pre-processing segmentation, feature extraction, tracking, and classification. Currently, this extension can be used with various image processing tools such as BioFormats, SCFIO, ImageJ, Omero, and SciJava [19].

### 2.3. Proposed Method

The proposed method presented in the study is based on three key aspects: (1) capturing image redundancy through a super pixel processing plane, (2) addressing image segmentation as a supervised super pixel classification problem, and (3) dynamically building the method by comparing multiple image enhancement techniques and classification algorithms to configure the final segmentation method. The outline of the proposed method is shown in Figure 3, which consists of three phases: pre-processing, feature extraction, and machine learning. In the pre-processing phase, original TMA images are enhanced using three different techniques, and simple linear iterative clustering (SLIC) is applied to obtain the super pixel images. In the feature extraction phase, 69 texture features and 1 class variable are extracted from the labeled images. In the machine learning phase, the data set is classified using five different algorithms, and the classified super pixels are regrouped to obtain the segmented images.

Pre-processing: The TMA images were initially very large, and it was found that they had a very high computational load without providing many benefits against a possible reduction in their dimension. Then, evaluating the available hardware resources, it was decided to scale them to 25%, leaving each image with a resolution of approximately 1500 × 1500 pixels. The best one was selected for the segmentation phase. These techniques were applied separately, producing three new sets of TMA images, one for each transformation.Image normalization: Normalizing an image changes the range of pixel intensity values in order to obtain greater consistency within the data set. In this case, the TMA images were normalized by applying the spatial contrast enhancement method.Histogram equalization: Histogram equalization is a nonlinear transformation technique that modifies the intensity value of pixels by distributing it over the entire spectrum, which produces a more constant histogram.Histogram matching: When an image’s histogram matches the specified histogram, it is known as histogram matching [18]. When a mathematical function or another image’s histogram is matched to an input image’s own histogram, the operation is called histogram matching. It is very useful for comparing and contrasting images. A valid comparison can be made if the histograms of the two photos in question are similar.SLIC super pixel construction: Groups of pixels that share features like color, brightness, or texture are known as super pixels. In addition to offering an easy starting point from which to compute image characteristics, they also considerably reduce the complexity of subsequent image processing operations. Histopathological pictures have shown it to be effective [20].Experimentation: The SLIC method was applied to the original (untransformed), normalized, and histogram equalization TMA images using ImageJ with the jSLIC plugin [20], which corresponds to a computationally more efficient variation of SLIC but retains the original method. To begin with, jSLIC is basically governed by two parameters, which are:
○Start grid size: the initial average size of super pixels (30 by default)○Regularization: degree of deformation of the estimated super pixels. The range is from 0 for very elastic super pixels to 1 for almost square super pixels (0.20 by default).


The application of jSLIC to the set of TMA images produced, as a result, super pixel images containing a label for each super pixel. The purpose of them is to point out the area of each super pixel which was necessary for the next phase of feature extraction.

Feature extraction: This stage consisted of obtaining a representation of the image (in the form of data) that would help discriminate between the different classes (tumor, non-tumor, and fundus). According to *Jurmeister*, et al. [21] the performance of a CAD system depends more on the extraction and selection of features than on the classification method.First-order statistics: The variance and other pixel neighborhood associations are not taken into account when computing statistics for first-order texture measures. A technique for texture analysis based on histograms examines how an image’s intensity concentrations—represented visually as a histogram—change over time. Moments such as mean, variance, spread, root mean square or mean energy, entropy, skewness, and kurtosis are among the most common characteristics of a distribution. A texture measure is a change in gray levels near a pixel, according to *Fernández-Carrobles* et al. [22].Tamura Characteristics: In *Tamura* et al. [23], six textural features that take advantage of human visual perception are introduced: roughness, roughness, contrast, directionality, linearity, and regularity. *Fernández-Carrobles* et al. [22] demonstrated the effectiveness of these visual characteristics, helping classifiers achieve success rates of up to 97% on histopathological images. The contrast and directionality features were selected as they were considered “more relevant” to obtain a better representation of the TMA images.Extracted features: For the problem addressed, it was determined that the texture characteristics were the most appropriate: 23 texture features were selected, divided into 17 first-order statistics and 6 Tamura-type features.Decomposition of RGB space: Because the original TMA images were in RGB space, it was necessary to decompose them into their three channels R (red), G (green), and B (blue), as a requirement for the calculation of the different types of features. In the end, a total of 69 features were obtained for each super pixel. For this, KNIME was used with the Splitter node to separate the image into channels and the Image Segment Features node to extract the selected features from each channel.Class assignment: In this step, the class variable (tumor, non-tumor, or background) was created and assigned to each feature vector (super pixel). For this, super pixel images were used, together with label images. The procedure consisted of taking the super pixel image by traversing each super pixel and interposing it over its corresponding area within the label image. This, in turn, has information on the class to which each pixel belongs, for which the number of pixels belonging to each class is counted through a voting system, and finally assigned to the super pixel the class that has the most votes. In other words, the class of the majority region to which the super pixel belongs is assigned.Segmentation: This stage consisted of the construction and comparison of five supervised classification models, of which the best was chosen and then used in the segmentation of the TMA images. First, five algorithms were selected from among the most relevant as evidenced in state of the art; then, once the models were trained for comparison, their performance was evaluated, and finally, the best segmentation algorithm was applied.

#### 2.3.1. Classification Algorithms

The following supervised classification algorithms were selected: Random Forest, Support Vector Machines (*SVM*), LogiBoost, J48, and BayesNet. Using the KNIME environment that Weka version 3.7 has integrated.

Random Forest: Breiman [24] introduced this method, which is currently one of the best and most often used. Using Random Forests is a mixture of prediction trees where each tree is dependent on the values of a random vector that is tested individually and with the same distribution for each one. A significant change from bagging is the construction and averaging of a large number of uncorrelated trees.Support Vector Machines (SVM): In huge data sets, the effectiveness of Sequential Minimal Optimization stands out. Training a support vector classifier is made easier by using Platt’s sequential minimum optimization method [25]. With this approach, no values are left out, and the nominal properties are converted from decimal to binary. All default attributes are also normalized. (In such a scenario, the output coefficients are based on the normalized data, not the original data, which is critical for evaluating the classifier).LogitBoost: It is the Weka implementation of the additive logistic regression algorithm introduced [26]. It can handle a wide range of issues and perform classification using a regression approach similar to the base learner. This algorithm was chosen over other boosting algorithms because of its superior performance in this case.J48: A pruned, or unpruned decision tree can be generated by the open-source implementation (Weka) of Quinlan’s C4.5 algorithm [25,26]. A statistical classifier is a type of tree. Information entropy is used to generate decision trees in C4.5, based on the training data.BayesNet: In other words, it is a Naive Bayes probability-classifier implementation. There are nodes and links in an ordered acyclic graph and conditional probability diagrams in this model [25,26]. The performance of this classifier was superior to other Bayesian-type classifiers.

#### 2.3.2. Evaluation Metrics

According to the problem dealt with, the approach used for training and testing varies a little with respect to the traditional one in supervised learning, since here not only the rows of the dataset must be classified (which in this case correspond to super pixels) but also needs to present the final segmentation of the image. Therefore, each time the method was tested with at least one TMA image and trained with the rest. The metrics used in the studies seen in art are diverse [27]. Of these, two of the most frequent have been selected. The confusion matrix of a binary classification evaluation must be understood before discussing these measures in depth (Table 1). In the training set, the class labels can only have one of two values: positive or negative. True positives (TP) and true negatives (TN) are the cases that a classifier properly predicts as positive or negative. False positives (FPs) and false negatives (FNs) are both terms for cases that were erroneously categorized (FNs).

True positives and negatives are counted per class in multiclass classification rather than in binary classification because there is no generic positive or negative class.

F-measure (F1-score): If you are asked to sort items into a class, the accuracy (also known as positive predictive value (PPV)) of a class is equal to one divided by the total number of positives. Classified as positive are those things.


(1)
PPV=TP/TP+FP


It is defined as the number of true positives divided by the number of items that genuinely belong to the positive class (that is, the sum of true positives and false negatives, which are situations that were not labeled as belonging to the positive class but should have been).
(2)TPR=TP/TP+FN

A weighted average of accuracy and recall can be used to interpret the F-measure in binary classification statistical analysis, with the best F1 score of 1 and the worst score of 0. The mean of the scores is used to get this value. When the two numbers are near, the harmonic mean, which is more commonly the square of the geometric mean divided by the arithmetic mean, is the answer.
(3)$$F_{1}=2\times \frac{PPV.TPR}{PPV+TPR}=\frac{2TP}{2TP+FP+FN}$$

It is possible to critique the F-score (Equation (3)) because of its bias as an evaluation metric because it does not take into consideration the number of true negatives (TN). Despite this, it is frequently used to evaluate the performance of categorization models.

Accuracy: There are many ways to quantify accuracy in binary classification tests, and precision is one of the most commonly utilized. In other words, the precision of a test measures the proportion of real outcomes (including real positives and real negatives) to the total number of instances investigated.


(4)
$$ACC=\frac{TP+TN}{TP+TN+FP+FN}$$


The precision gives an overview (at the multiclass level) of the performance of the built model and, as seen in Equation (4), the precision does take into account the number of true negatives TN.

Cross-validation: Using cross-validation, you can make sure that your statistical findings are not skewed by the fact that training and test data are separate. K-iteration cross-validation divides the sample data into K subgroups for each iteration of the test. Data from one subset is utilized for testing, while data from the remaining K subsets are used for training. Repeating cross-validation with each of the potential test data subsets for K iterations is the standard procedure. In the end, the arithmetic mean of each iteration’s results is calculated to get the final result. Then, to statistically evaluate and guarantee the results, the 10-fold cross-validation technique was used: in each iteration, the training set was made up of nine images and the test set, one image. This, finally, facilitates the comparison of the results obtained with others from state-of-the-art, since many of the methods used this technique to validate their models.

Ground truth annotation: The ground truth is the procedure that allows evaluating the precision of the segmentation of the CAD system when comparing it with the manual segmentation made by the experts. According to Gertych et al. [28], when ground truth (tagged images) is present, segmentation can be measured in terms of accuracy (hit rate) and robustness. The precision on its part reflects the accuracy of the segmentation with respect to the marked images.

The ground truth was performed using the same class assignment procedure, taking into account that the purpose of the procedure is the same: assign a class to the super pixel based on its marked image.

Segmented images: Finally, based on the classification, segmented images were constructed, which are similar to the label images but with the difference that they were produced by the CAD system. It must be remembered that initially the image was decomposed into its super pixels, which were classified. Therefore, it was necessary to maintain a column with the position information of each super pixel in the image within the classification data set. Thus, after the classification, the information of that column (which was not taken into account by the classifier) was used to place each super pixel in its position, grouping them according to their class.

### 2.4. Pre-Processing

This phase began by reducing the dimension of the images with a scaling of 25%, then three image enhancement methods were applied: normalization, histogram matching, and histogram equalization, with which three improved TMA image data sets were obtained; as an example, only one is presented for each transformation as show in Figure 4.

Standardization, Histogram Equalization, and Histogram Matching.

## 3. Results

### 3.1. SLIC Super Pixels

The jSLIC method was applied on the set of initial TMA images using the parameters, and the result can be seen in Figure 5a,b.

The previous image is made up of 3567 super pixels, so at this scale it is difficult to clearly detail the adjustment of the super pixels on the histopathological structures. However, below Figure 5c,d, an approach is made to visualize this aspect in more detail. Figure 5c,d shows the good adjustment of the super pixels to the three different regions. On the left is healthy tissue, on the right (brown) is the tumor tissue, and below (light gray) in the background. The yellow lines indicate the limits of the super pixels and their adjustment to the structures.

### 3.2. Extracted Features

In this phase, initially 23 characteristics of the first-order Tamura and statistical types were selected; however, as the image was in RGB format, for the calculation of characteristics, it had to be divided into its three channels. Therefore, 69 characteristics were obtained for each super pixel which was assigned their respective classes. Finally, a data set of 70 variables (69 predictors and one class) was obtained with a total of 37,279 cases (super pixels). The distribution of classes is shown below (Figure 6).

### 3.3. Classification

The experimentation strategy in the previous phases of pre-processing, super pixel construction, and feature extraction consisted of testing different parameter configurations to find the most optimal ones. This phase also maintained this strategy by comparing five different state-of-the-art algorithms. These were tested on the three enhanced TMA image sets.

The metrics for evaluating the performance of the classifiers are precision (accuracy) and F-measure (F-score), and 10-fold cross-validation was performed. These metrics are an indicator of the goodness of the classifier at a general level in multiclass problems; although sometimes one or the other is used, it was necessary for this study to take both to facilitate the comparison with the reference methods.

#### 3.3.1. Results in Normalized Images

The precision obtained per image (sub-sample) for each algorithm using the normalized images as input. In this case, the algorithm that shows the best result is Random Forest, followed by SVM with similar behavior. The image best predicted by all the algorithms was nine, which indicates that in this iteration, the training obtained a good explanatory capacity from the other images. On the other hand, image 10 has the highest variance in its classification. Although it is one of the best classified by Random Forest and SVM (close to 90% accuracy), it is the worst classified by J48 and LogitBoost. The average performance for each algorithm is presented in Figure 7. The goodness measures indicate that the algorithm that obtained the best performance is Random Forest with an accuracy and F-measure of 83%, followed by SVM, LogitBoost, J48, and BayesNet.

#### 3.3.2. Histogram Equalization Imaging Results

For this data set, the Random Forest algorithm presents a precision per image superior to the other algorithms, even exceeding 90% (Figure 7). Figure 8 is the worst classified by all algorithms. Finally, BayesNet is the algorithm that shows the lowest performance for most images. Regarding the general average performance (Figure 8), the Random Forest algorithm stands out with an accuracy of 83% and an F-measure of 84%. In this case, second place was obtained by LogitBoost and then by SVM, J48, and BayesNet.

#### 3.3.3. Image Results of Histogram Matching

For the histogram matching data set, the superior performance of Random Forest can be seen in most of the images, except image two, where BayesNet obtains greater precision; the best-classified image (by most algorithms) is nine with an accuracy of 90% obtained by SVM together with Random Forest and Figure 8 is the poorest classified. The goodness measures (Figure 9) indicate that Random Forest is the algorithm with the highest average performance for this data set, with an accuracy and F-measure of 83%; it far exceeds SVM and LogitBoost (same result), and finally J48 and BayesNet.

#### 3.3.4. Comparative Analysis

The best measures of goodness (highlighted in Figure 7, Figure 8 and Figure 9) indicated that the algorithm that presented the highest average performance during the tests was Random Forest as show in Figure 10. Therefore, it was the algorithm selected to configure the final segmentation method.

In addition to selecting the best algorithm, it was necessary to select the best image enhancement technique. This was the histogram equalization that, together with Random Forest, is the one that obtains the best classification result. This technique is followed by histogram matching and finally normalization with quite similar results.

### 3.4. Segmented Images

This study used the segmentation task as a super pixel classification problem. A comparative analysis determined that histogram equalization was the best image enhancement technique, and that the classification algorithm was Random Forest; together, they obtained an average precision of 83.4% and an F-measurement of 84.4%. With this configuration, the final segmentation was obtained, and the results obtained with the proposed method are presented below (Figure 11). In the images in columns (b) and (c), the colors correspond to red: tumor, green: non-tumor, and white: background.

It should be mentioned that a certain degree of (human) error is implicit in marked images; however, said marking is taken as true to have a reference to compare the automatically segmented images. Now, when making this comparison (Figure 11), it can be seen that the background (in white) is well-differentiated from the tissue for most of the images. However, when looking at image two it can be seen that non-tumor parts are classified As a background, if the initial image (a) is examined in more detail, it can be seen that the experts marked these parts as non-tumor. This could be subject to review by the experts to confirm whether or not the automatic segmentation is correct. However, human error is known to be a factor present within tagged images. This is not a major problem and is quite common in manual-dealing tasks.

Regarding the tumor area, which is the most relevant, it is observed that in general, the method is correct in the segmentation, distinguishing the main tumor regions (red color). However, it is also evident above all in the segmentation of the images (c) four, five, and eight that there are small areas classified as a tumor that are actually non-tumor. Unlike the true tumor region, which is distinguished by being grouped into larger areas, these small areas are characterized by being small and distributed throughout the non-tumor area. The best-segmented image is nine (c); this is very close to the manually marked image (b). The most notable differences are parts of the tumor region misclassified as non-tumor. Observing this aspect in more detail, it is believed that many of these areas correspond to a non-tumor area. Therefore, the CAD system could have been successful in this part of its segmentation. On the other hand, the worst segmented image is number eight, where the shapes of the tumor regions are not clearly seen but are lost among small areas poorly classified as non-tumor areas. This would indicate that the most confusing classes for the classifier are tumor and non-tumor.

From the model accuracy (Figure 12), the accuracy of the model is increased by an increase in the accuracy of the period. It means that precision completely depends on the model’s iteration or training size.

The cross-entropy function regulates the loss of a model and evaluates the model loss. From model loss (Figure 13), it is seen that the model loss lowers with an increase in time.

## 4. Discussion

Initially, three image enhancement techniques were compared, of which histogram equalization significantly improved the final segmentation of the CAD system. Although image enhancement techniques were created to improve human visual interpretation, this study shows that they are also effective in improving interpretation by computational systems. A very interesting aspect is that the histogram equalization, unlike the other two techniques, does not “normalize” at the data set level but rather works independently on each image and, although its results are good, it would not be possible to know the result of trying one of them. In images with a difference too far from the rest of the set it is evident that the work with super pixel SLICs through which one goes from a pixel classification plane to one of the super pixels has been effective in capturing the redundancy of the TMA images as well as achieving a good adjustment to their different histopathological structures. It should be noted that this process could be improved in the future by optimizing the parameters of the SLIC method.

On the other hand, it has been observed that although the super pixels capture the redundancy of the image, the use of very large super pixels does not help to obtain good texture characteristics. On the contrary, it introduces a greater error in the ground truth process. The estimation of the error (and its minimization) between the images marked by the experts and the segmented ones is an aspect to consider in the future.

The extracted texture features were appropriated to build descriptors that helped classification algorithms to better discriminate between classes. Although the constructed data set has a good size when compared to similar studies, it is evident that it is relatively small. One possibility to increase the descriptive power could be to include a greater number of TMA images in addition to using an automatic variable selection method, given that in many cases the performance of the model depends more on the selected characteristics than on the classification algorithm.

Regarding the classification, it was observed that the algorithm that presented the best performance for the three data sets of the three image improvement transformations was Random Forest, reaching an average accuracy of 83.4% and an F-measurement of 84.4%. This result is quite close to that obtained by similar studies for prostate cancer [29,30]. It was observed that for the studies with success rates above 90%, a common factor was the use of large data sets (more than 100 images), which translates into more robust models with a broader explanatory capacity. Another aspect to take into account is that not all the studies used TMA images; some used traditional image slices, so their models only classified tumor and non-tumor areas. The use of TMA images gave the present study a significant advantage as it ensures a common and more reproducible framework at a scientific level.

In general, the proposed method performs a good segmentation; the resulting images are quite similar to the marked images. Although small areas of the different zones are incorrectly segmented, it can be said that there is the possibility that some of these areas misclassified are in fact false positives, which implies that with a more exhaustive marking by the experts, the evaluation of the error can be optimized. The performance of the model can be improved.

## 5. Conclusions

Lung cancer is the most common cancer in the world. In this work, an approximation to its detection was made, approaching the problem of segmentation of tumor tissue and non-tumor tissue on TMA histopathological images through the development of a prototype of a computer-assisted diagnostic system that allowed segmenting the tumor areas, non-tumor, and fundus with an average accuracy of 83.4% and an F-measurement of 84.4%.

A very Interesting finding was that some misclassified tissue regions could be mainly due to experts’ non-detailed (omitted) annotations; the CAD system discovered tumor regions that were not marked by the experts, which constituted it in “a second diagnostic opinion”.

A promising method has been developed to effectively approach the problem of histopathological analysis, which also considerably reduces the time to diagnosis. The proposed method has future potential for quantifying lung cancer; thus, opening a path to knowing its status more precisely, improving its estimation, and predicting progress.

Super pixel SLIC-based pre-processing captures image redundancy reducing processing complexity. Texture features extracted from super pixels produce good class descriptors. This finally optimizes the classification stage.

The need for CAD systems in histopathology is pressing. It contributes to a method of early detection of lung cancer, considering first that histopathological imaging is the gold standard in cancer detection and second that most of the work done in CAD for lung cancer has focused on radiology and has shown that this area is not heading in the early detection of cancer.

Although this study was framed in detecting lung cancer, in theory, a method has been built that could extend its application to other types of cancer since most of them are diagnosed using histopathological imaging. Therefore, the use of TMA images constitutes an advantage that contributes to achieving this goal in the future.

### Future Work

The main future objective is to quantify tumor and non-tumor areas and offer pathologists indicators of lung cancer estimation and progress. It is suggested to experiment with the parameters of the super pixels, both with the size and degree of deformation, to achieve more efficient super pixels. Extract new features to increase the descriptive power of the dataset and use an automatic variable selection method as a dimensionality reduction strategy. Human error is implicit in marking tagged images; it is important to know and estimate it. A proposed alternative would be to mark the images by more than one expert and measure the inter-observer variance. The imaging market needs to be refined by experts. The set of images is relatively small; it is proposed to increase the number of marked images to increase the model’s explanatory capacity. As the last proposal, the possibility of adding a new post-processing phase to debug those incorrectly classified super pixels is proposed.

## Figures and Tables

**Figure 1 jpm-13-00388-f001:**
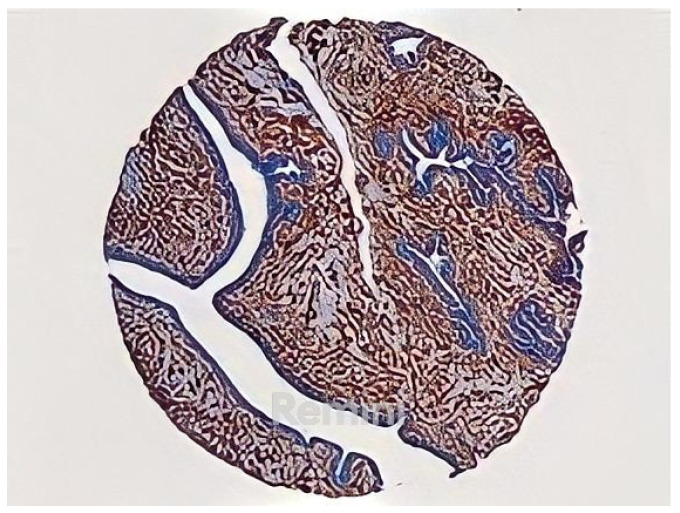
Digitized TMA slide.

**Figure 2 jpm-13-00388-f002:**
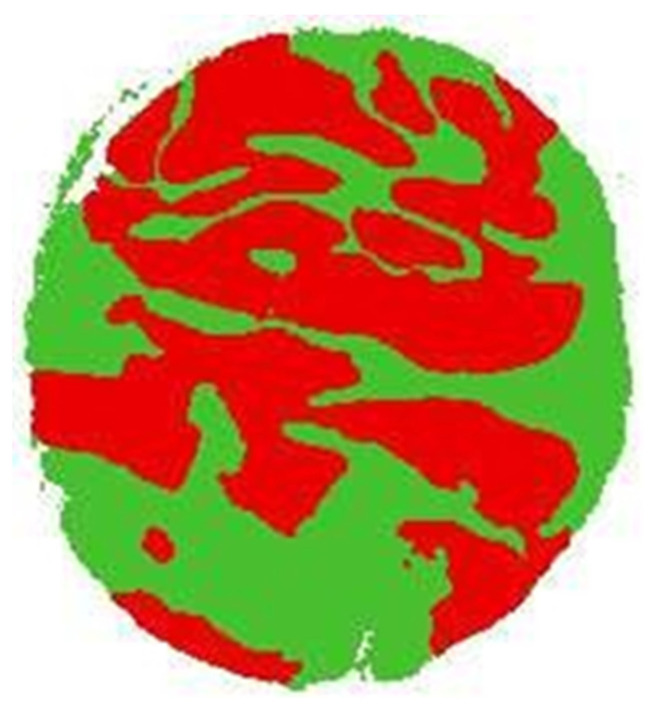
Labeled TMA, zones: red-tumor, green-non-tumor, and white-background.

**Figure 3 jpm-13-00388-f003:**
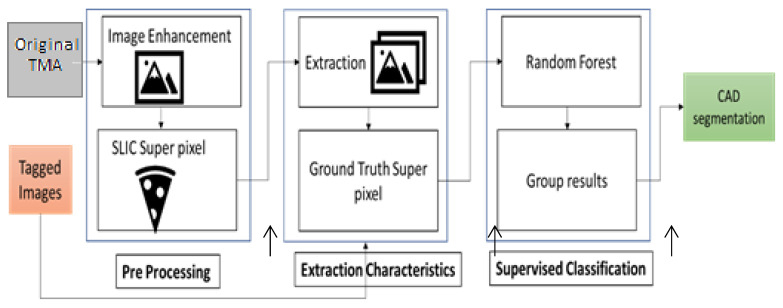
Scheme of the proposed method.

**Figure 4 jpm-13-00388-f004:**

Transformed by normalization (**b**), from the original image (**a**); transformed by equalization of the histogram (**d**), from the original image (**c**); transformed by histogram matching (**f**), from the original image (**e**).

**Figure 5 jpm-13-00388-f005:**
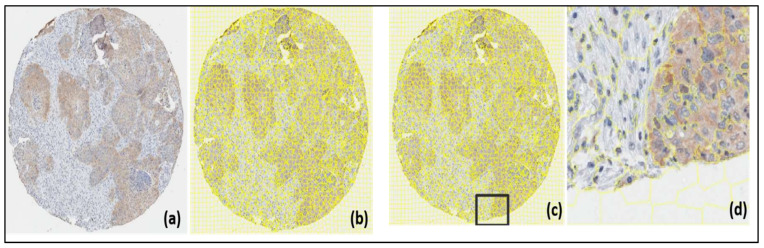
Segmentation by super pixels (**b**), from the original image (**a**); super pixel segmentation (**d**), cropped version of (**c**).

**Figure 6 jpm-13-00388-f006:**
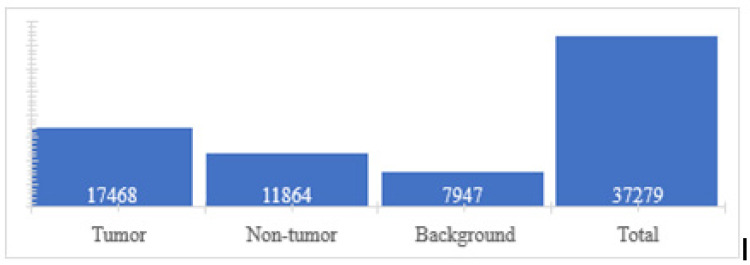
Number of features per class.

**Figure 7 jpm-13-00388-f007:**
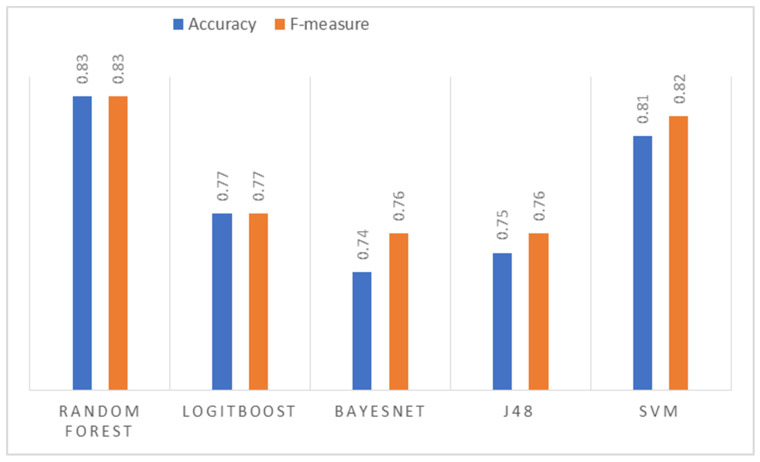
Comparison of metrics—average performance.

**Figure 8 jpm-13-00388-f008:**
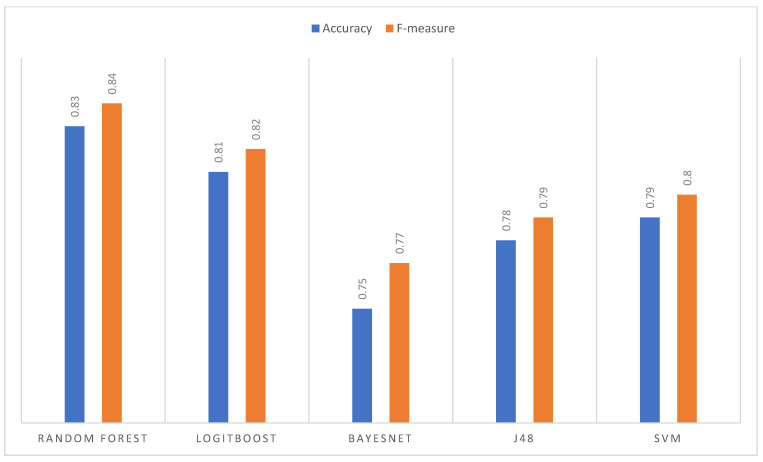
Comparison of metrics (average performance).

**Figure 9 jpm-13-00388-f009:**
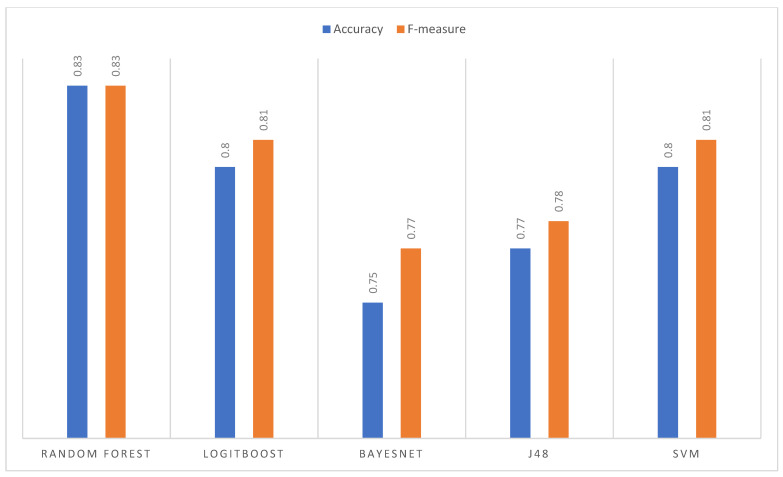
Comparison of metrics (average performance).

**Figure 10 jpm-13-00388-f010:**
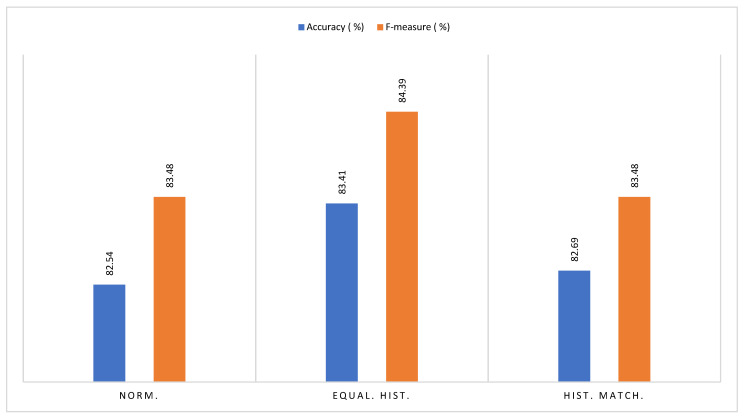
Average Random Forest performance per enhanced imagery data set.

**Figure 11 jpm-13-00388-f011:**
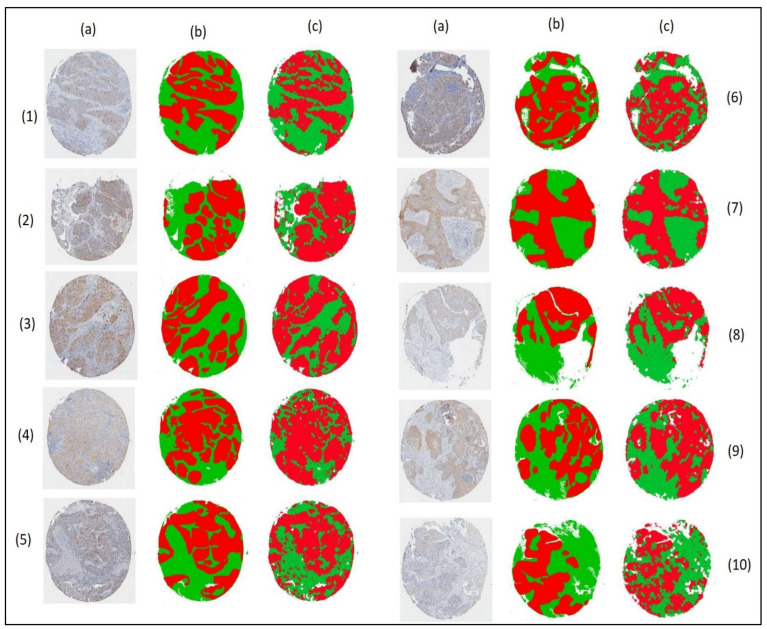
Segmentation result of the proposed method. Column (**a**) is the original image, (**b**) the image marked by the expert and (**c**) the CAD automatic segmentation.

**Figure 12 jpm-13-00388-f012:**
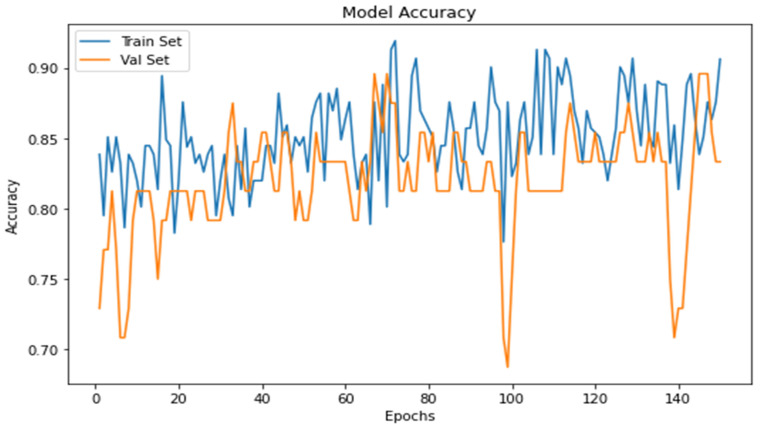
Graph between model accuracy and epochs.

**Figure 13 jpm-13-00388-f013:**
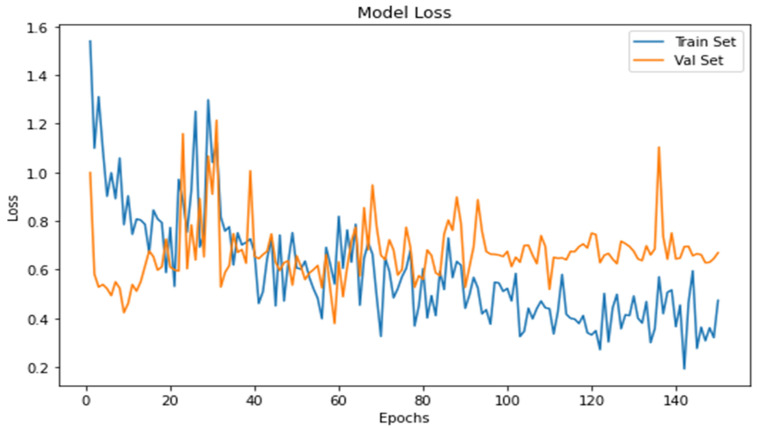
Graph between model loss and epochs.

**Table 1 jpm-13-00388-t001:** Binary classification confusion matrix.

	Positive (P)	Negative (N)
Actual True	TP	FN
Actual False	FP	TN

## Data Availability

Not applicable.

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
