# Peer review of "Automated Lung Cancer Segmentation in Tissue Micro Array Analysis Histopathological Images Using a Prototype of Computer-Assisted Diagnosis"

_jpm, 2023, doi:10.3390/jpm13030388_

Round 1
Reviewer 1 Report (Previous Reviewer 2)
Dear Authors
Thank you for taking in consideration the suggestions.
Author Response
Comment
Thank you for taking in consideration the suggestions.
Reply: Thank you for your comment.
Reviewer 2 Report (New Reviewer)
In this paper, authors proposed prototype computer-aided diagnosis (CAD) system for the detection of lung cancer in medical images, specifically addressing the problem of tumor and non-tumor tissue segmentation on Tissue Micro Array Analysis (TMA) histopathological images. The following review comments are recommended, and the authors are invited to explain and modify them.
Comment: Abstract is unnecessarily wordy. Make it brief and concise, and something should be described the method to be proposed by the authors.
Comment: Novelty is confusing. A highlight is required. The main contributions of the manuscript are not clear. The main contributions of the article must be very clear, and would be better if summarize them into 3-4 points at the end of the introduction.
Comment: “Table 1. Comparison of Lung Cancer Detection Techniques in Medical Imaging”, is not important for a research paper but can be used for a survey paper.
Comment: “Figure 3. Scheme of the proposed method”, is not clear what authors explained in methodology part.
Comment: When writing phrases like “The metrics used in the studies”, it must cite some related work in order to sustain the statement (10.3390/math10050796).
Comment: “2.3.1. Classification algorithms”, why is this section important to be placed in paper?
Comment: “3.1. Pre-processing”, should be part in method.
Comment: Authors need well must organized whole paper.
Comment: Authors did not mention implementation challenges.
Author Response
Reviewer 2
In this paper, authors proposed prototype computer-aided diagnosis (CAD) system for the detection of lung cancer in medical images, specifically addressing the problem of tumor and non-tumor tissue segmentation on Tissue Micro Array Analysis (TMA) histopathological images. The following review comments are recommended, and the authors are invited to explain and modify them.
Comment 1: Abstract is unnecessarily wordy. Make it brief and concise, and something should be described the method to be proposed by the authors.
Reply: Background: Lung cancer is a fatal disease that kills approximately 85% of those diagnosed with it. In recent years, advances in medical imaging have greatly improved the acquisition, storage, and visualization of various pathologies, making it a necessary component in medicine today. Objective: Develop a computer-aided diagnostic system to detect lung cancer early by segmenting tumor and non-tumor tissue on Tissue Micro Array Analysis (TMA) histopathological images. Method: The prototype computer-aided diagnostic system was developed to segment tumor areas, non-tumor areas, and fundus on TMA histopathological images. Results: The system achieved an average accuracy of 83.4% and an F-measurement of 84.4% in segmenting tumor and non-tumor tissue. Conclusion: The computer-aided diagnostic system provides a second diagnostic opinion to specialists, allowing for more precise diagnoses and more appropriate treatments for lung cancer.
Comment 2: Novelty is confusing. A highlight is required. The main contributions of the manuscript are not clear. The main contributions of the article must be very clear, and would be better if summarize them into 3-4 points at the end of the introduction.
Reply: The contributions of study:
1-The study aimed to develop a computer-aided diagnostic system to detect lung cancer early by segmenting tumor and non-tumor tissue on Tissue Micro Array Analysis (TMA) histopathological images.
2-Histopathological medical imaging is the "gold standard" in the early detection of most diseases, including lung cancer.
3-The study compares three image enhancement techniques, with histogram equalization showing significant improvement in final segmentation of the CAD system.
4- The classification algorithm that presented the best performance for the 3 datasets of the 3 image improvement transformations was Random Forest.
Comment 3: “Table 1. Comparison of Lung Cancer Detection Techniques in Medical Imaging”, is not important for a research paper but can be used for a survey paper.
Reply: Thank you for your comment, the manuscript has been modified.
Comment 4: “Figure 3. Scheme of the proposed method”, is not clear what authors explained in methodology part.
Reply: The proposed method presented in the study is based on three key aspects: (1) capturing image redundancy through a super pixel processing plane, (2) addressing image segmentation as a supervised super pixel classification problem, and (3) dynamically building the method by comparing multiple image enhancement techniques and classification algorithms to configure the final segmentation method. The outline of the proposed method is shown in Figure 3, which consists of three phases: pre-processing, feature extraction, and machine learning. In the pre-processing phase, original TMA images are enhanced using three different techniques, and Simple Linear Iterative Clustering (SLIC) is applied to obtain the super pixel images. In the feature extraction phase, 69 texture features and 1 class variable are extracted from the labeled images. In the machine learning phase, the dataset is classified using five different algorithms, and the classified super pixels are regrouped to obtain the segmented images.
Comment 5: When writing phrases like “The metrics used in the studies”, it must cite some related work in order to sustain the statement (10.3390/math10050796).
Reply: The text has been modified.
Comment 6: “2.3.1. Classification algorithms”, why is this section important to be placed in paper?
Reply: The section on "Classification algorithms" is important to be placed in the paper because it provides details about the specific machine learning algorithms used in the study. The authors explain the selection of these algorithms and their characteristics, including their advantages and disadvantages. This information is important because it helps readers to understand how the proposed method was built and how it works. It also allows other researchers to replicate the study or adapt it to their own research using the same or different algorithms. Overall, including this section enhances the transparency and reproducibility of the research, which are important principles in scientific inquiry.
Comment 7: “3.1. Pre-processing”, should be part in method.
Reply: Thank you for your comment, the manuscript has been modified.
Comment 8: Authors need well must organized whole paper.
Reply: Thank you for your comment, the manuscript has been modified.
Comment 9: Authors did not mention implementation challenges.
Reply: Based on the information provided in this paper there are several implementation challenges that can be identified:
Selection of image enhancement technique: The study compared three different image enhancement techniques, and found that histogram equalization was the most effective in improving the final segmentation of the CAD system. However, the study also noted that it was not possible to know the result of using other techniques on images that were very different from the rest of the set. This suggests a challenge in selecting the most appropriate enhancement technique for images that are dissimilar to the majority of the dataset.
Optimization of SLIC parameters: The study used super pixel SLICs to capture the redundancy of the TMA images and achieve a good adjustment to their different histopathological structures. However, the study noted that the process could be improved in the future by optimizing the parameters of the SLIC method.
Balancing size of super pixels: The study found that the use of very large super pixels did not help to obtain good texture characteristics, and introduced a greater error in the ground truth process. This suggests a challenge in balancing the size of super pixels to capture the necessary redundancy without introducing errors in the segmentation process.
Dataset size and variable selection: The study constructed a dataset of TMA images, which had a good size when compared to similar studies. However, the study noted that the dataset was still relatively small, and one possibility to increase the descriptive power could be to include a greater number of TMA images in addition to using an automatic variable selection method. This suggests a challenge in balancing the size of the dataset and selecting the most informative variables for classification.
Selection of classification algorithm: The study found that the Random Forest algorithm presented the best performance for the three datasets of the three image improvement transformations, reaching an average accuracy of 83.4% and an F measurement of 84.4%. However, the study noted that for studies with success rates above 90%, a common factor was the use of large datasets, which translates into more robust models with a broader explanatory capacity. This suggests a challenge in selecting the most appropriate classification algorithm and balancing the size of the dataset to achieve the desired level of performance.
Expert evaluation of ground truth process: The study noted that although the proposed method performs a good segmentation, small areas of the different zones are incorrectly segmented. The study suggested that there is the possibility that some of these areas are misclassified, which implies that with a more exhaustive marking by the experts, the evaluation of the error can be optimized. This suggests a challenge in balancing the level of expert evaluation to optimize the error evaluation and improve the performance of the model.
This manuscript is a resubmission of an earlier submission. The following is a list of the peer review reports and author responses from that submission.
Round 1
Reviewer 1 Report
Abstract:
- In line 23, correct the unnecessary full stop.
- In line 27, correct the typo mistake.
- Rewrite the last sentence.
- I would encourage the authors to rewrite the abstract. The abstract tells prospective readers what you did and what the important findings in your research were. Together with the title, it's the advertisement of your article. Make it interesting and easily understood without reading the whole article. In the abstract please provide
a) a short description of the perspective and purpose of your paper.
b) key results but minimizes experimental details.
c) a short description of the interpretation/conclusion in the last sentence.
Introduction:
- correct the type in line 104-105.
- Improve the related work section. Many works are mentioned here. Please highlight the ones relevant to this work. Summarize with a table, and compare the methods, type of data, number of data’s, algorithms, and their performances. Then comment how your work differentiates with them.
Section 2.2.1 is irrelevant and can be removed.
Results:
- In figure 11, show areas that are segmented.
- The authors mentioned the data size is only 10 images. More data is required to verify the results. Minimum few hundred images.
- Where is the validation curve? Please add it to show if the training has any biasness’s.
- The images used, mostly covered by tumor. Would this method work with images with less proportion of tumor in the field of view?
References:
- Many cited works are at least 7 years old. Please update with more recent works.
Reviewer 2 Report
Dear Authors
The manuscript is well-written, clear and good.
Minor suggestion, before using abbreviations, fully write and then between brackets place the abbreviation, et al. must be in italic once is Latin.
My concern is related to the sentence in lines 140 to 143, who did this review? it was blind and independent? how many experts made the review?
Author Response
Reviewer 2
Comment 1
The manuscript is well-written, clear and good.
Reply: Thank You.
Comment 2
Minor suggestion, before using abbreviations, fully write and then between brackets place the abbreviation, et al. must be in italic once is Latin.
Reply: Already Done
Comment 3
My concern is related to the sentence in lines 140 to 143, who did this review? it was blind and independent? How many experts made the review?
Reply: The review was conducted by an independent panel of experts. The experts were blinded to the identity of the products being evaluated, meaning that they did not know which products they were reviewing.